# Certified Monotonic Neural Networks

**Xingchao Liu**
Department of Computer Science
University of Texas at Austin
Austin, TX 78712
xcliu@utexas.edu

**Xing Han**
Department of Electrical and Computer Engineering
University of Texas at Austin
Austin, TX 78712
aaronhan223@utexas.edu

**Na Zhang**
Tsinghua University
zhangna@pbcsf.tsinghua.edu.cn

**Qiang Liu**
Department of Computer Science
University of Texas at Austin
Austin, TX 78712
lqiang@cs.utexas.edu

## Abstract

Learning monotonic models with respect to a subset of the inputs is a desirable feature to effectively address the fairness, interpretability, and generalization issues in practice. Existing methods for learning monotonic neural networks either require specifically designed model structures to ensure monotonicity, which can be too restrictive/complicated, or enforce monotonicity by adjusting the learning process, which cannot provably guarantee the learned model is monotonic on selected features. In this work, we propose to certify the monotonicity of the general piece-wise linear neural networks by solving a mixed integer linear programming problem. This provides a new general approach for learning monotonic neural networks with arbitrary model structures. Our method allows us to train neural networks with heuristic monotonicity regularizations, and we can gradually increase the regularization magnitude until the learned network is certified monotonic. Compared to prior works, our method does not require human-designed constraints on the weight space and also yields more accurate approximation. Empirical studies on various datasets demonstrate the efficiency of our approach over the state-of-the-art methods, such as Deep Lattice Networks [34].

## 1 Introduction

Monotonicity with respect to certain inputs is a desirable property of the machine learning (ML) predictions in many practical applications [e.g., 17, 28, 11, 9, 10, 6]. For real-world scenarios with fairness or security concerns, model predictions that violate monotonicity could be considered unacceptable. For example, when using ML to predict admission decisions, it may seem unfair to select student X over student Y, if Y has a higher score than X, while all other aspects of the two are identical. A similar problem can arise when applying ML in many other areas, such as loan application, criminal judgment, and recruitment. In addition to the fairness and security concerns, incorporating the monotonic property into the ML models can also help improve their interpretability, especially for the deep neural networks [22]. Last but not least, enforcing monotonicity could increase the generalization ability of the model and hence the accuracy of the predictions [10, 34], if the enforced monotonicity pattern is consistent with the underlying truth.

While incorporating monotonicity constraints has been widely studied for the traditional machine learning and statistical models for decades [e.g., 9, 8, 5, 27, 2, 21], the current challenge is how to incorporate monotonicity into complex neural networks effectively and flexibly. Generally, existing approaches for learning monotonic neural networks can be categorized into two groups:

1) *Hand-designed Monotonic Architectures.* A popular approach is to design special neural architectures that guarantee monotonicity by construction [e.g., 2, 7, 10, 34]. Unfortunately, these designed monotonic architectures can be very restrictive or complex, and are typically difficult to implement in practice. A further review of this line of work is provided at the end of Section 1.

2) *Heuristic Monotonic Regularization.* An alternative line of work focuses on enforcing monotonicity for an arbitrary, off-the-shelf neural network by training with a heuristically designed regularization (e.g., by penalizing negative gradients on the data) [13]. While this approach is more flexible and easier to implement compared to the former method, it cannot provably ensure that the learned models would produce the desired monotonic response on selected features. As a result, the monotonicity constraint can be violated on some data, which may lead to costly results when deployed to solve real-world tasks.

Obviously, each line of the existing methods has its pros and cons. In this work, we propose a new paradigm for learning monotonic functions that can gain the best of both worlds: leveraging arbitrary neural architectures and provably ensuring monotonicity of the learned models. The key of our approach is an optimization-based technique for mathematically verifying, or rejecting, the monotonicity of an arbitrary piece-wise linear (e.g., ReLU) neural network. In this way, we transform the monotonicity verification into a mixed integer linear programming (MILP) problem that can be solved by powerful off-the-shelf techniques. Equipped with our monotonicity verification technique, we can learn monotonic networks by training the networks with heuristic monotonicity regularizations and gradually increasing the regularization magnitude until it passes the monotonicity verification. Empirically, we show that our method is able to learn more flexible partially monotonic functions on various challenging datasets and achieve higher test accuracy than the existing approaches with best performance, including the recent Deep Lattice Network [34]. We also demonstrate the use of monotonic constraints for learning interpretable convolutional networks.

**Related works:** As we have categorized the existing work into two groups earlier, here we further summarize some concrete examples that are most relevant to our work. A simple approach to obtain monotonic neural networks is to constrain the weights on the variables to be non-negative [2]. This, however, yields a very restrictive subset of monotonic functions (e.g., ReLU networks with non-negative weights are always convex) and does not perform well in practice. Another classical monotonic architecture is the Min-Max network [7], which forms a universal approximation of monotonic functions theoretically, but does not work well in practice. Deep Lattice Network (DLN) [34] exploits a special class of function, an ensemble of lattices [10], as a differentiable component of neural network. DLN requires a large number of parameters to obtain good performance.

Moreover, the monotonicity verification that we propose admits a new form of verification problem of the ReLU networks that has not been explored before, which is, verifying a property of the gradients on the whole input domain. Existing work has investigated verification problems that include evaluating robustness against adversarial attack [31, 25, 35], and computing the reachable set of a network [3, 20]. Compared with these problems , verifying monotonicity casts a more significant challenge because it is a global property on the whole domain rather than a local neighborhood (this is true even for the individual monotonicity that we introduce in Section 3.1). Given its practical importance, we hope our work can motivate further exploration in this direction.

## 2   Monotonicity in Machine Learning

We present the concept of monotonicity and discuss its importance in practical applications. In particular, we introduce a form of adversarial attacking that exploits the non-monotonicity in problems for which fairness plays a significant role.

**Monotonic and Partial Monotonic Functions** Formally, let $f(\boldsymbol{x})$ be a neural network mapping from an input space $\mathcal{X}$ to $\mathbb{R}$. In this work, we mainly consider the case when $\mathcal{X}$ is a rectangle region in $\mathbb{R}^d$, i.e., $\mathcal{X} = \otimes_{i=1}^d [l_i, u_i]$. Assume the input $\boldsymbol{x}$ is partitioned into $\boldsymbol{x} = [\boldsymbol{x}_\alpha, \boldsymbol{x}_{\neg\alpha}]$, where $\alpha$ is a subset of $[1, \ldots, d]$ and $\neg\alpha$ its complement, and $\boldsymbol{x}_\alpha := [x_i : i \in \alpha]$ is the corresponding sub-vector of $\boldsymbol{x}$. Denote the space of $\boldsymbol{x}_\alpha$ and $\boldsymbol{x}_{\neg\alpha}$ by $\mathcal{X}_\alpha = \otimes_{i\in\alpha}[l_i, u_i]$ and $\mathcal{X}_{\neg\alpha} := \otimes_{i\in\neg\alpha}[l_i, u_i]$ respectively. We say that $f$ is (partially) monotonic w.r.t $\boldsymbol{x}_\alpha$ if

$$f(\boldsymbol{x}_\alpha, \boldsymbol{x}_{\neg\alpha}) \le f(\boldsymbol{x}'_\alpha, \boldsymbol{x}_{\neg\alpha}), \quad \forall \boldsymbol{x}_\alpha \le \boldsymbol{x}'_\alpha, \quad \forall \boldsymbol{x}_\alpha, \boldsymbol{x}'_\alpha \in \mathcal{X}_\alpha, \ \boldsymbol{x}_{\neg\alpha} \in \mathcal{X}_{\neg\alpha}, \tag{1}$$

where $\boldsymbol{x}_\alpha \le \boldsymbol{x}'_\alpha$ denotes the inequality for all the elements, that is, $x_i \le x'_i$ for all $i \in \alpha$.

**Individual Monotonicity and Monotonicity Attacking** In fields where fairness and security are of critical importance, it is highly desirable to enforce monotonicity over certain features in the deployed ML models [17, 28, 11]. Otherwise, the system may be subject to attacks that exploit the non-monotonicity within it. Consider, for example, a program for predicting a product price (e.g., house) based on the product features. Let $\boldsymbol{x}_\alpha$ be the features that people naturally expect to be monotonic (such as the quantity or quality of the product). For a product with feature $\boldsymbol{x} = [\boldsymbol{x}_\alpha, \boldsymbol{x}_{\neg\alpha}]$, if the function is not monotonic w.r.t. $\boldsymbol{x}_\alpha$, then we can find another testing example $\hat{\boldsymbol{x}} = [\hat{\boldsymbol{x}}_\alpha, \hat{\boldsymbol{x}}_{\neg\alpha}]$, which satisfies

$$f(\hat{\boldsymbol{x}}) > f(\boldsymbol{x}), \ s.t. \ \hat{\boldsymbol{x}}_\alpha \leq \boldsymbol{x}_\alpha, \ \ \hat{\boldsymbol{x}}_{\neg\alpha} = \boldsymbol{x}_{\neg\alpha}. \tag{2}$$

In other words, while $\hat{\boldsymbol{x}}$ has the same values on the non-monotonic features with $\boldsymbol{x}$, and smaller values on the monontonic features than $\boldsymbol{x}$, $f(\hat{\boldsymbol{x}})$ is larger than $f(\boldsymbol{x})$. If such case is possible, the fairness of the system would be cast in doubt. Addressing this kind of problems is critical for many real-world scenarios such as criminal judgment, loan applications, as well as hiring/administration decisions. In light of this, we call $f$ to be *individually monotonic* on $\boldsymbol{x}$ if there exists no adversarial example as described in (2).

The non-monotonicity is hard to detect through a simple sanity check, unless the model is monotonic by construction. For example, Figure 1 shows a data instance $\boldsymbol{x}$ we found on COMPAS [16], a recidivism risk score dataset. In this example, a trained neural network is monotonic with respect to the monotonic features (i.e., $f([x_i, \boldsymbol{x}_{\neg i}])$ w.r.t. each $x_i$ with $\boldsymbol{x}_{\neg i}$ fixed on the instance), but there exists an adversarial example $\hat{\boldsymbol{x}}$ that violates the monotonicity in the sense of (2). In this case, checking the monotonicity requires us to eliminate all the combinations of features on the input domain. To do so, we need a principled optimization framework, which can eliminate the existence of any possible monotonicity violations.

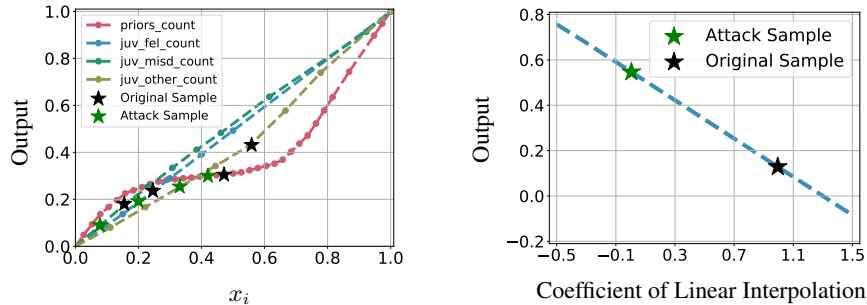

Figure 1: If monotonicity is not strictly enforced, there may exist misleading cases when the model appears to be monotonic for each individual feature with a simple sanity check, such as visualizing the 1D slice plot of the individual features (Left), but there may exist an adversarial example that violates the monotonicity in the sense of (2) (Right). Here, we trained a two-layer ReLU network with a heuristic monotonicity regularization on the COMPAS dataset, which has 4 monotonic features out of 13. The stars in the left figure indicates the value of each monotonic feature. The right figure shows the linear slice ($\boldsymbol{x} + \alpha(\hat{\boldsymbol{x}} - \boldsymbol{x})$ where $\alpha$ is the coefficient of linear interpolation) from the data point $\boldsymbol{x}$ to its adversarial example $\hat{\boldsymbol{x}}$.

# 3 Learning Certified Monotonic Networks

In this section, we introduce our main method for learning certified monotonic networks. We start by discussing how to verify individual monotonicity or otherwise find monotonic adversarial examples (Section 3.1), followed by verifying the global monotonicity on the whole domain (Section 3.2). We then discuss our learning method (Section 3.3), and extend the monotonicity verification to the multiple layer neural networks (Section 3.4).

## 3.1 Certifying Individual Monotonicity

For a given data point $\boldsymbol{x}$ and a model $f$, we want to either verify the non-existence of any monotonicity adversarial examples, or otherwise detect all such adversarial examples if they exist. Detecting a

monotonicity adversarial example can be framed into the following optimization problem:

$$\hat{\boldsymbol{x}}_\alpha^* = \arg\max_{\boldsymbol{x}' \in \mathcal{X}} f(\boldsymbol{x}'_\alpha, \boldsymbol{x}_{\neg\alpha}) \quad s.t. \quad \boldsymbol{x}'_\alpha \le \boldsymbol{x}_\alpha, \quad \boldsymbol{x}'_{\neg\alpha} = \boldsymbol{x}_{\neg\alpha}. \tag{3}$$

If $f(\hat{\boldsymbol{x}}^*) > f(\boldsymbol{x})$, then $\hat{\boldsymbol{x}}^*$ is a monotonic adversarial example. Otherwise, no monotonicity attacking is possible. Eq (3) amounts to solving a challenging non-convex optimization problem. To tackle it, we first note that most neural networks use piece-wise linear activation functions (ReLU, leaky ReLU, etc.). This fact implies that the optimization can be framed into a mixed integer linear programming (MILP) problem, which can be solved by leveraging the powerful off-the-shelf techniques. Specifically, let $f(\boldsymbol{x})$ be a two-layer ReLU network,

$$f(\boldsymbol{x}) = \sum_{i=1}^n a_i \text{ReLU}(\boldsymbol{w}_i^\top \boldsymbol{x} + b_i). \tag{4}$$

The ReLU activation, $\text{ReLU}(\boldsymbol{w}_i^\top \boldsymbol{x} + b_i)$, can be rewritten into a set of mixed integer linear constraints as follows:

$$y_i = \text{ReLU}(\boldsymbol{w}_i^\top \boldsymbol{x} + b_i) \quad \Leftrightarrow \quad y_i \in \mathcal{C}(\boldsymbol{x}, \boldsymbol{w}_i, \boldsymbol{b}_i), \tag{5}$$

$$\text{where} \quad \mathcal{C}(\boldsymbol{x}, \boldsymbol{w}_i, \boldsymbol{b}_i) = \left\{ y \;\middle|\; \begin{array}{ll} y \ge 0, & y \le u_i z, \quad z \in \{0, 1\} \\ y \ge \boldsymbol{w}_i^\top \boldsymbol{x} + b_i, & y \le \boldsymbol{w}_i^\top \boldsymbol{x} + b_i - l_i(1 - z) \end{array} \right\},$$

in which $z$ is a binary variable that indicates whether ReLU is activated or not, and $u_i = \sup_{\boldsymbol{x} \in \mathcal{X}} \{\boldsymbol{w}_i^\top \boldsymbol{x} + b_i\}$ and $l_i = \inf_{\boldsymbol{x} \in \mathcal{X}} \{\boldsymbol{w}_i^\top \boldsymbol{x} + b_i\}$ are the maximum and minimum values of the output respectively. Both $u_i$ and $l_i$ can be calculated easily when $\mathcal{X}$ is a rectangular interval in $\mathbb{R}^d$. For example, when $\mathcal{X} = [0, 1]^d$, we have $u_i = \text{ReLU}(\boldsymbol{w}_i)^\top \mathbf{1} + b_i$, where $\mathbf{1}$ denotes the vector of all ones. Eq (5) is an important characterization of the ReLU that has been widely used for other purposes [31, 12, 3, 26].

Following these, we are now ready to frame the optimization in (3) as

$$\max_{\boldsymbol{x}'} \sum_{i=1}^n a_i y_i, \quad s.t. \quad \boldsymbol{x}'_\alpha \le \boldsymbol{x}_\alpha, \quad \boldsymbol{x}'_{\neg\alpha} = \boldsymbol{x}_{\neg\alpha}, \quad y_i \in \mathcal{C}(\boldsymbol{x}, \boldsymbol{w}_i, b_i), \quad \forall i \in [n].$$

It is straightforward to develop a similar formulation for networks with more layers. Besides, our method can also be extended to neural networks with smooth activation functions by upper bounding the smooth activation functions with piece-wise linear functions; see Appendix B.2 for details.

## 3.2 Monotonicity Verification

In addition to the individual monotonicity around a given point $\boldsymbol{x}$, it is important to check the global monotonicity for all the points in the input domain as well. It turns out that we can also address this problem through an optimization approach. For a differentiable function $f$, it is monotonic w.r.t. $\boldsymbol{x}_\alpha$ on $\mathcal{X}$ if and only if $\partial_{x_\ell} f(\boldsymbol{x}) \ge 0$ for all $\ell \in \alpha, \boldsymbol{x} \in \mathcal{X}$. We can check this by solving

$$U_\alpha := \min_{\boldsymbol{x}, \ell \in \alpha} \{\partial_{x_\ell} f(\boldsymbol{x}), \; \boldsymbol{x} \in \mathcal{X}\} \tag{6}$$

If $U_\alpha \ge 0$, then monotonicity is verified. Again, we can turn this optimization into a MILP for the ReLU networks. Consider the ReLU network in (4). Its gradient equals,

$$\partial_{x_\ell} f(\boldsymbol{x}) = \sum_{i=1}^n \mathbb{I}(\boldsymbol{w}_i^\top \boldsymbol{x} + b_i \ge 0) a_i w_{i,\ell}. \tag{7}$$

Following the same spirit as the previous section, we are able to transform the indicator function $\mathbb{I}(\boldsymbol{w}_i^\top \boldsymbol{x} + b_i \ge 0)$ into a mixed integer linear constraint,

$$z_i = \mathbb{I}(\boldsymbol{w}_i^\top \boldsymbol{x} + b_i \ge 0) \quad \Leftrightarrow \quad z_i \in \mathcal{G}(\boldsymbol{x}, \boldsymbol{w}_i, \boldsymbol{b}_i), \tag{8}$$

$$\text{where} \quad \mathcal{G}(\boldsymbol{x}, \boldsymbol{w}_i, \boldsymbol{b}_i) = \left\{ z_i \;\middle|\; z_i \in \{0, 1\}, \quad \boldsymbol{w}_i^\top \boldsymbol{x} + b_i \le u_i z_i, \quad \boldsymbol{w}_i^\top \boldsymbol{x} + b_i \ge l_i(1 - z_i) \right\}. \tag{9}$$

Here, $u_i$ and $l_i$ are defined as before. One can easily verify the equivalence: if $\boldsymbol{w}_i^\top \boldsymbol{x} + b_i \ge 0$, then $z_i$ must be one, because $\boldsymbol{w}_i^\top \boldsymbol{x} + b_i \le u_i z_i$; if $\boldsymbol{w}_i^\top \boldsymbol{x} + b_i \le 0$, then $z_i$ must be zero, because $\boldsymbol{w}_i^\top \boldsymbol{x} + b_i \ge l_i(1 - z_i)$.

Therefore, we can turn (6) into a MILP:

$$U_\alpha = \min_{\boldsymbol{x}, \ell \in \alpha} \left\{ \sum_{i=1}^n a_i w_{i,\ell} z_i \quad s.t. \quad z_i \in \mathcal{G}(\boldsymbol{x}, \boldsymbol{w}_i, b_i), \quad \boldsymbol{x} \in \mathcal{X} \right\}. \tag{10}$$

**MILP Solvers:** There exists a number of off-the-shelf MILP solvers, such as GLPK library [23] and Gurobi [14]. These solvers are based on branch-and-bound methods, accompanied with abundant of heuristics to accelerate the solving process. Due to the NP nature of MILP [3], it is impractical to obtain exact solution when the number of integers is too large (e.g., 1000). Fortunately, most MILP solvers are *anytime*, in that they can stop under a given budget to provide a lower bound of the optimal value (in case, a lower bound of $U_\alpha$). Then it verifies the monotonicity without solving the problem exactly. A simple example of lower bound can be obtained by linear relaxation, which has already been widely used in verification problems associated with neural networks [33, 35]. It has been an active research area to develop tighter lower bounds than linear relaxation, including using tighter constraints [1] or smarter branching strategies [3]. Since these techniques are available in off-the-shelf solvers, we do not further discuss them here.

## 3.3 Learning Certified Monotonic Neural Networks

We now introduce our simple procedure for learning monotonic neural networks with verification. Our learning algorithm works by training a typical network with a data-driving monotonicity regularization, and gradually increase the regularization magnitude until the network passes the monotonicity verification in (6). Precisely, it alternates between the following two steps:

**Step 1:** Training a neural network $f$ by

$$\min_f \mathcal{L}(f) + \lambda R(f), \qquad \text{where} \qquad R(f) = \mathbb{E}_{x \sim \mathrm{Uni}(\mathcal{X})} \Big[ \sum_{\ell \in \alpha} \max(0, -\partial_{x_\ell} f(\boldsymbol{x}))^2 \Big], \qquad (11)$$

where $\mathcal{L}(f)$ is the typical training loss, and $R(f)$ is a penalty that characterizes the violation of monotonicity; here $\lambda$ is the corresponding coefficient and $\mathrm{Uni}(\mathcal{X})$ denotes the uniform distribution on $\mathcal{X}$. $R(f)$ can be defined heuristically in other ways. $R(f) = 0$ implies that $f$ is monotonic w.r.t. $\boldsymbol{x}_\alpha$, but it has to be computationally efficient. For example, $U_\alpha$ in (6) is not suitable because it is too computationally expensive to be evaluated at each iteration of training.

The exact value of $R(f)$ is intractable, and we approximate it by drawing samples of size 1024 uniformly from the input domain during iterations of the gradient descent. Note that the samples we draw vary from iteration to iteration. By the theory of stochastic gradient descent, we can expect to minimize the object function well at convergence. Also, training NNs requires more than thousands of steps, therefore the overall size of samples can well cover the input domain. In practice, we use a modified regularization $R(f) = \mathbb{E}_{x \sim \mathrm{Uni}(\mathcal{X})} \Big[ \sum_{\ell \in \alpha} \max(b, -\partial_{x_\ell} f(\boldsymbol{x}))^2 \Big]$, where $b$ is a small positive constant, because we find the original version will always lead to a $U_\alpha$ that is slightly smaller than zero.

**Step 2:** Calculate $U_\alpha$ or a lower bound of it. If it is sufficient to verify that $U_\alpha \geq 0$, then $f$ is monotonic and the algorithm terminates, otherwise, increase $\lambda$ and repeat step 1.

This training pipeline requires no special architecture design or constraints on the weight space. Though optimizing $R(f)$ involves computation of second order derivative, we found it can be effectively computed in modern deep learning frameworks. The main concern is the computational time of the monotonicity verification, which is discussed in Section 3.4.

## 3.4 Extension to Deep Neural Networks

Although it is possible to directly extend the verification approach above to networks with more than two layers by formulating a corresponding MILP, the resulting optimization may include a large number of integer variables, making the problem intractable. See Appendix B.1 for detailed discussion. In this section, we discuss a more practical approach for learning and verifying monotonic deep networks by decomposing the network into a stack of two-layer networks and then verifying their monotonicity separately.

Assume $f \colon \mathcal{X} \to \mathbb{R}$ is a deep ReLU network with an even number $2K$ of layers (otherwise, we can add an identity layer on the top and fix its weights during training). We decompose the network into a composition of two-layer networks:

$$f(\boldsymbol{x}) = f_{2K:2K-1} \circ \cdots \circ f_{4:3} \circ f_{2:1}(\boldsymbol{x}),$$

where $f_{2k:2k-1}$ denotes the composition of the $2k$-th and $(2k-1)$-th layers of $f$. Therefore, a sufficient condition for $f$ to be monotonic is that all $f_{2k:2k-1}, \forall k = 1, \ldots, K$ are monotonic, each of

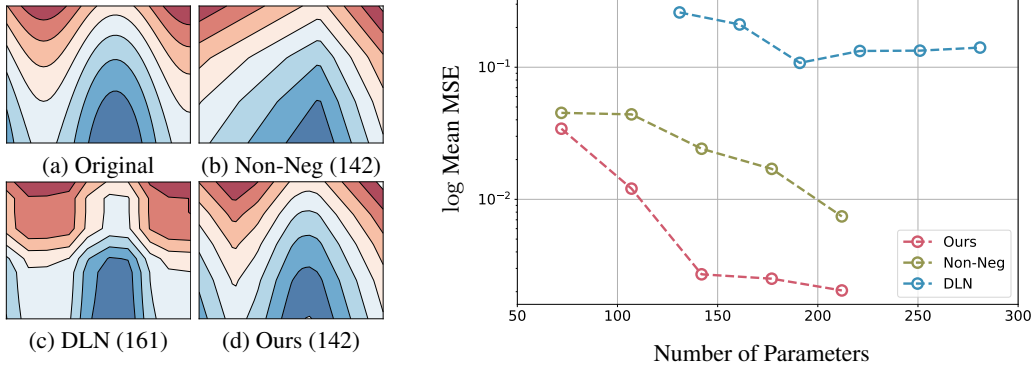

(a) Original     (b) Non-Neg (142)

(c) DLN (161)     (d) Ours (142)

Figure 2: We test Deep Lattice Network (DLN) [34], networks with non-negative weights (Non-Neg) [2], and our method on fitting a family of 2D functions: $f(x, y) = a \sin(x/25\pi) + b (x-0.5)^3 + c \exp(y) + y^2, a, b, c \in \{0.3, 0.6, 1.0\}$. **Left:** The fitting result when $a = 1.0, b = 1.0, c = 1.0$. The number in the parenthesis refers to the number of parameters of the model. Our method fits the original function best. **Right:** We test the above methods on fitting all the 27 functions with different number of parameters. We averaged the mean-square-error (MSE) of all 27 runs. Our method yields better performance than the other methods.

which can be verified separately using our method in Section 3.2. We normalize the input feature to $[0, 1]$. To address the change of input domain across the layers, we derive the corresponding upper and lower bound $u_i$ and $l_i$ from $u_{i-1}$ and $l_{i-1}$. We can evaluate all the $u_i$ and $l_i$ in a recursive manner.

Obviously, the layer-wise approach may not be able to verify the monotonicity in the case when $f$ is monotonic, but not all the $f_{2k:2k-1}$ layers are. To address this problem, we explicitly enforce the monotonicity of all $f_{2k:2k-1}$ during training, so that they can be easily verified using the layer-wise approach. Specifically, we introduce the following regularization during training:

$$\tilde{R}(f) = \sum_{k=1}^{K} R(f_{2k:2k-1}), \tag{12}$$

where $R$ can be defined as (11). See in Algorithm 1 in Appendix for the detailed procedure.

The idea of using two-layer (vs. one-layer) decomposition allows us to benefit from the extended representation power of deep networks without significant increase of computational cost. Note that two-layer networks form universal approximation in the space of bounded continuous functions, and hence allows us to construct highly flexible approximation. If we instead decomposed the network into the stack of one-layer networks, the verification becomes simply checking the signs of the weights, which is much more restrictive.

## 4 Experiments

### 4.1 Comparison with Other Methods

We verify our method in various practical settings and datasets. Experiment results show that networks learned by our method can achieve higher test accuracy with fewer parameters, than the best-known algorithms for monotonic neural networks, including Min-Max Network [7] and Deep Lattice Network [34]. Our method also outperforms traditional monotonic methods, such as isotonic regression and monotonic XGBoost, in accuracy. We also demonstrate how to learn interpretable convolutional neural networks with monotonicity.

**Datasets:** Experiments are performed on 4 datasets: COMPAS [16], Blog Feedback Regression [4], Loan Defaulter[1], Chest X-ray[2]. *COMPAS* is a classification dataset with 13 features. 4 of them are

| Method | Parameters | Test Acc |
|---|---|---|
| Isotonic | N.A. | 67.6% |
| XGBoost [5] | N.A. | 68.5% ± 0.1% |
| Crystal [10] | 25840 | 66.3% ± 0.1% |
| DLN [34] | 31403 | 67.9% ± 0.3% |
| Min-Max Net [7] | 42000 | 67.8% ± 0.1% |
| Non-Neg-DNN | 23112 | 67.3% ± 0.9% |
| **Ours** | **23112** | **68.8% ± 0.2%** |

Table 1: Results on COMPAS

| Methods | Parameters | RMSE |
|---|---|---|
| Isotonic | N.A. | 0.203 |
| XGBoost [5] | N.A. | 0.176 ± 0.005 |
| Crystal [10] | 15840 | 0.164 ± 0.002 |
| DLN [34] | 27903 | 0.161 ± 0.001 |
| Min-Max Net [7] | 27700 | 0.163 ± 0.001 |
| Non-Neg-DNN | 8492 | 0.168 ± 0.001 |
| **Ours** | **8492** | **0.158 ± 0.001** |

Table 2: Results on Blog Feedback

| Methods | Parameters | Test Acc |
|---|---|---|
| Isotonic | N.A. | 62.1% |
| XGBoost [5] | N.A. | 63.7% ± 0.1% |
| Crystal [10] | 16940 | 65.0% ± 0.1% |
| DLN [34] | 29949 | 65.1% ± 0.2% |
| Min-Max Net [7] | 29000 | 64.9% ± 0.1% |
| Non-Neg-DNN | 8502 | 65.1% ± 0.1% |
| **Ours** | **8502** | **65.2% ± 0.1%** |

Table 3: Results on Loan Defaulter

| Methods | Parameters | Test Acc |
|---|---|---|
| XGBoost [5] | N.A. | 64.4% ± 0.4% |
| Crystal [10] | 26540 | 65.3% ± 0.1% |
| DLN [34] | 39949 | 65.4% ± 0.1% |
| Min-Max Net [7] | 35130 | 64.3% ± 0.6% |
| Non-Neg-DNN | 12792 | 64.7% ± 1.6% |
| Ours w/o E-to-E | 12792 | 62.3% ± 0.2% |
| **Ours** | **12792** | **66.3% ± 1.0%** |

Table 4: Results on Chest X-Ray. 'w/o E-to-E' means the weights in the pretrained feature extractor are frozen during training.

monotonic. *Blog Feedback* is a regression dataset with 276 features. 8 of the features are monotonic. *Loan Defaulter* is a classification dataset with 28 features. 5 of them are monotonic. The dataset includes half a million data points. *Chest X-Ray* is a classification dataset with 4 tabular features and an image. 2 of the tabular features are monotonic. All the images are resized to $224 \times 224$. For each dataset, we pick 20% of the training data as the validation set. More details can be found in appendix.

**Methods for Comparison:** We compare our method with six methods that can generate partially monotonic models. *Isotonic Regression*: a deterministic method for monotonic regression [9]. *XGBoost*: a popular algorithm based on gradient boosting decision tree [5]. *Crystal*: an algorithm using ensemble of lattices [10]. *Deep Lattice Network (DLN)*: a deep network with ensemble of lattices layer [34]. *Non-Neg-DNN*: deep neural networks with non-negative weights. *Min-Max Net*: a classical three-layer network with one linear layer, one min-pooling layer, and one max-pooling layer [7]. For Non-Neg-DNN, we use the same structure as our method.

**Hyper-parameter Configuration:** We use cross-entropy loss for classification problems, and mean-square-error for regression problems. 20% of the training data is used as the validation set. All the methods use the same training set and validation set. We validate the number of neurons in each layer and the depth of the network. Adam [18] optimizer is used for optimization. For solving the MILP problems, we adopt Gurobi v9.0.1 [14], which is an efficient commercial solver. We initialize the coefficient of monotonicity regularization $\lambda = 1$, and multiply $\lambda$ by 10 every time $\lambda$ needs amplification. The default learning rate is $5e - 3$. When $\lambda$ is large, $5e - 3$ may cause training failure. In this case, we decrease the learning rate until training successes. Our method is implemented with PyTorch [24]. All the results are averaged over 3 runs. The code is publicly available[3].

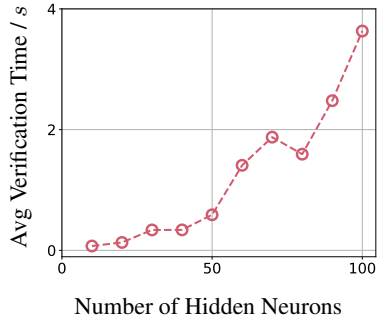

Figure 3: Verification time w.r.t. number of hidden neurons.

[3]https://github.com/gnobitab/CertifiedMonotonicNetwork

**Our Method Learns Smaller, More Accurate Monotonic Networks:** The results on the dataset above are summarized in Table 1, 2, 3, and 4. It shows that our method tends to outperform all the other methods in terms of test accuracy, and learns networks with fewer parameters. Note that because our method use only typical neural architectures, it is also easier to train and use in practice. All we need is adding the monotonicity regularization in the loss function.

**Our Method Learns Non-trivial Sign Combinations:** Some neural networks, such as those with all non-negative weights, can be trivially verified to be monotonic. More generally, a neural network can be verified to be monotonic by just reading the sign of the weights (call this *sign verification*) if the product of the weights of all the paths connecting the monotonic features to the outputs are positive. Let us take a two-layer ReLU network, $f = W_2 ReLU(W_1 x)$, for example. Because $ReLU(\cdot)$ is a monotonically increasing function, we can verify the monotonicity of the network if all the elements in the matrix $W_2 W_1$ is non-negative without our MILP formulation. Each element in the matrix is a multiplication of the weights on a path connecting the input to the output, hence we call such paths *non-negative/negative paths*. As shown in Table. 5 and Fig. 4.1, our method tends to learn neural networks that cannot be trivially verified by sign verification, suggesting that it learns in a richer space of monotonic functions. $A, B, C, D$ refer to four different networks, with different structures and trained on different datasets.

**Computational Time for Monotonicity Verification:** Because our monotonicity verification involves solving MILP problems, we evaluate the time cost of two-layer verification in Fig. 3. All the results are averaged over 3 networks trained with different random seeds on COMPAS. The verification can be done in less than 4 seconds with 100 neurons in the first layer. Our computer has 48 cores and 192GB memory.

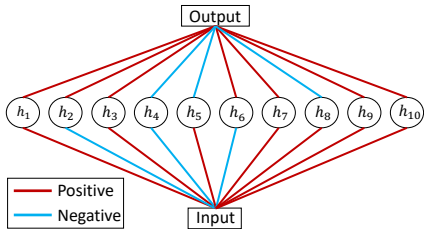

| Net | # of Paths | # of Negative Paths |
|-----|-----------|---------------------|
| A | 100,000 | 42,972 |
| B | 400 | 113 |
| C | 160 | 47 |
| D | 50000 | 21344 |

Figure 4: Weights learned of a two-layer monotonic net. $h_2, h_5, h_6, h_8$ are on negative paths.

Table 5: Statistics of negative paths

## 4.2 Learning Interpretable Neurons with Monotonic Constraints

Enforcing monotonicity provides a natural tool for enhancing the interpretablity of neural networks, but has not been widely explored in the literature with very few exceptions [22]. Here, we show an example of learning interpretable convolutional networks via monotonicity. We use MNIST [19] and consider binary classification between pairs of digits (denoted by $C_1$ and $C_2$). The network consists of three convolutional layers to extract the features of the images. The extracted features are fed into two neurons (denoted by $A$ and $B$), and are then processed by a hidden layer, obtaining the the class probabilities $P(C_1)$ and $P(C_2)$ after the softmax operation; see Fig. 5(a). To enforce interpretability, we add monotonic constraints during training such that $P(C_1)$ (resp. $P(C_2)$) is monotonically increasing to the output of neuron $A$ (resp. $B$), and is monotonically decreasing to neuron $B$ (resp. $A$). We adopt the previous training and verification pipeline, and the convolutional layers are also trained in an end-to-end manner. We visualize the gradient map of the output of neuron $A$ w.r.t. the input image via SmoothGrad [29]. As we show in Fig. 5(c), in the monotonic network, the top pixels in the gradient map identifies the most essential patterns for classification in that removing them turns the images into the opposite class visually.

## 4.3 Monotonicity Increases Adversarial Robustness

Interpretability of a model is considered to be deeply related to its robustness against adversarial attacks [32, 15, 36, 30]. People believe higher interpretability indicates higher robustness. Here, we empirically show that our interpratable models, trained with monotonicity constraints, do have

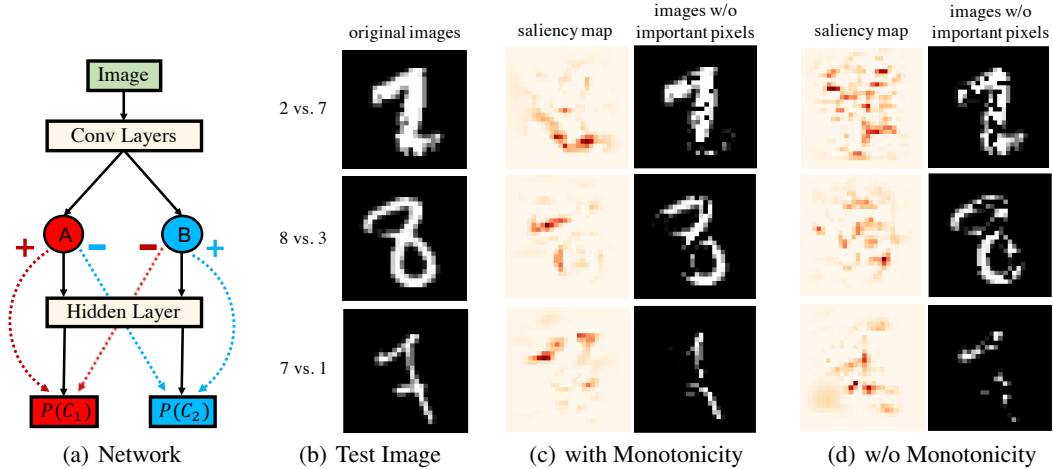

| | original images | saliency map | images w/o important pixels | saliency map | images w/o important pixels |

(a) Network　　　　(b) Test Image　　　　(c) with Monotonicity　　　　(d) w/o Monotonicity

Figure 5: (a) We train a neural network on MNIST with the constraint that $P(C_1)$ (resp. $P(C_2)$) is monotonically increasing w.r.t. neuron $A$ (resp. $B$), and monotonically decreasing w.r.t. neuron $B$ (resp. $A$). (b) Visualization of three binary classification tasks between two digits: 2 vs. 7 (1st row), 8 vs. 3 (2nd row), 7 vs. 1 (3rd row). We train the same network with and without monotonic constraints. (c) and (d) show the result when training the network with and without monotonic constraints, respectively. **Left column of (c) and (d):** The gradient heat map of neuron A, where higher value means the corresponding pixel has higher importance in predicting the image to be class $C_1$. **Right column of (c) and (d):** The image that we obtain by removing the most important pixels with the top 5% largest gradient values. We can see that in (c), in the monotonic network, removing the important pixels of a test image (such as the digit 2, 8, 7 in (b)) turns the image to the opposite class (e.g., 2 is turned to a 7 like image on the top row). In contrast, as shown in (d), removing the top-ranked pixels in the non-monotonic network makes little semantic change on the image.

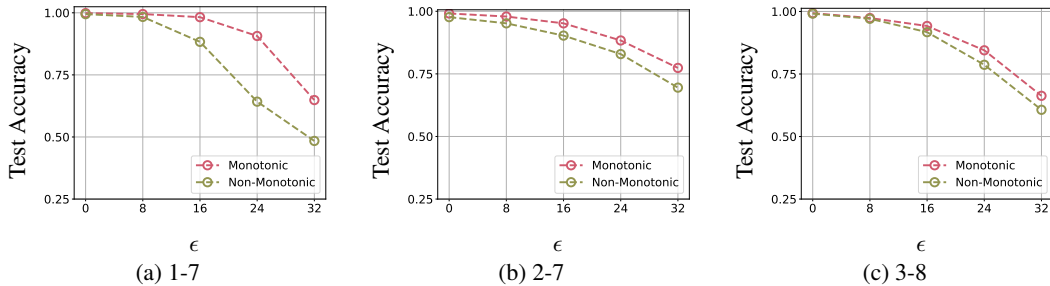

(a) 1-7　　　　　　　　(b) 2-7　　　　　　　　(c) 3-8

Figure 6: We perform PGD attack on the networks trained in Sec. 4.2, and test them on those binary classification problems. For clean images ($\epsilon = 0$), the test accuracy of the monotonic networks and the non-monotonic ones are almost the same. However, the monotonic networks show higher test accuracy over the non-monotonic counterparts under different magnitudes of adversarial attacks.

better performance under adversarial attacks. We take the trained convolutional neural networks in Sec. 4.2, and apply projected gradient descent (PGD) attack on the test images. We use a step size of $2/255$, and iterates for 30 steps to find the adversarial examples. We bound the difference between the adversarial image and the original image in a $\mathcal{L}_{\text{inf}}$ ball with radius $\epsilon$. A larger $\epsilon$ indicates a more significant attack. We show our results in Fig. 6.

## 5 Conclusions

We propose a verification-based framework for learning monotonic neural networks without specially designed model structures. In future work, we plan to investigate better verification methods to speed up, and to incorporate monotonicity into large modern convolutional neural networks to train interpretable networks.

**Broader Impact Statement:** Our method can simplify and improve the process of incorporating monotonic constraints in deep learning systems, which can potentially improve the fairness, security and interpretability of black-box deep models. Since it is a fundamental machine learning methodology, We do not foresee negative impact to the society implied by the algorithm directly.

**Funding Disclosure:** Work supported in part by NSF CAREER #1846421, SenSE #2037267, and EAGER #2041327. Xingchao Liu is supported in part by a funding from BP.

## Footnotes

[1]https://www.kaggle.com/wendykan/lending-club-loan-data

[2]https://www.kaggle.com/nih-chest-xrays/sample

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
