[Supplementary Material]

# A Details of the Experiments

## A.1 Details of the Datasets

Here we introduce the details of the datasets used in the experiments.

| Dataset | Task | Feature Dimension | Monotonic Features | # Training | # Test |
|---|---|---|---|---|---|
| COMPAS | Classification | 13 | 4 | 4937 | 1235 |
| Blog Feedback | Regression | 276 | 8 | 47302 | 6968 |
| Loan Defaulter | Classification | 28 | 5 | 418697 | 70212 |
| Chest X-Ray | Classification | 4 tabular + image | 2 | 4484 | 1122 |

Table 6: Dataset Summary

**COMPAS:** COMPAS [16] is a dataset containing the criminal records of 6,172 individuals arrested in Florida. The task is to predict whether the individual will commit a crime again in 2 years. The probability predicted by the system will be used as a risk score. We use 13 attributes for prediction. The risk score should be monotonically increasing w.r.t. four attributes, `number of prior adult convictions`, `number of juvenile felony`, `number of juvenile misdemeanor`, and `number of other convictions`.

**Blog Feedback Regression:** Blog Feedback [4] is a dataset containing 54,270 data points from blog posts. The raw HTML-documents of the blog posts were crawled and processed. The prediction task associated with the data is the prediction of the number of comments in the upcoming 24 hours. The feature of the dataset has 276 dimensions, and 8 attributes among them should be monotonically non-decreasing with the prediction. They are A51, A52, A53, A54, A56, A57, A58, A59. Please refer to the link [4] for the specific meanings of these attributes. Because outliers could dominate the MSE metric, we only use the data points with targets smaller than the 90th percentile.

**Prediction of Loan Defaulters:** Lending club loan data[5] contains complete loan data for all loans issued through 2007-2015 of several banks. Each data point is a 28-dimensional feature including the current loan status, latest payment information, and other additional features. The task is to predict loan defaulters given the feature vector. The possibility of loan default should be non-decreasing w.r.t. `number of public record bankruptcies`, `Debt-to-Income ratio`, and non-increasing w.r.t. `credit score`, `length of employment`, `annual income`.

**Chest X-Ray:** Without the constraints on structure, our method can easily go beyond tabular data. Chest X-ray exams are one of the most frequent and cost-effective medical imaging examinations available. NIH Chest X-ray Dataset[6] has 5606 X-ray images with disease labels and patient information. Hence, this dataset is a multi-modal dataset using both image and tabular data. We resize all the images to $224 \times 224$, and use a ResNet-18 pretrained on ImageNet as the feature extractor. The task is to predict whether a patient has chest disease or not. The possibility of chest disease is set to be non-decreasing to `age` and `number of follow-up examinations`. We did not count the parameters in the ResNet-18 feature extractor. The benefit of monotonic neural networks is that we can apply end-to-end training on the feature extractor. Other methods, including XGBoost, Crystal and DLN, cannot do end-to-end training. Hence, for these methods, we extract the features of the images using the pretrained ResNet-18, and train them using fixed image features without the ResNet-18 in the training pipeline.

## A.2 More Details in Implementation

To capture the non-monotonic relationship between the output and the non-monotonic features, we only impose monotonic constraints on half of the neurons in each $2k$-th layer. we cut off the connection (i.e. set the weights on the edges to zero) between the monotonic features and the other half of the neurons, so that removing the monotonic constraints will not change the monotonicity

**Algorithm 1** Training Monotonic Neural Network with Monotonic Verification
---
1: **Input:** A randomly initialized neural network $f$, dataset $\mathcal{D} = \{\boldsymbol{x}^{(i)}, y^{(i)}\}_{i=1}^{n}$ and the indices of monotonic features $I_m = \{m_1, m_2, \ldots, m_k\}$.
2: Set the coefficient of the monotonic regularization $\lambda = \lambda_0$.
3: Train $f$ with loss function $\mathcal{L}_{\mathcal{D}}(f) + \lambda R_{I_m}(f)$ till convergence.
4: **if** $f_{2k:2k-1}$ passes monotonic verification for $\forall k = 1, 2, \ldots, K$ **then**
5:     Return monotonic neural network $f$
6: **else**
7:     Increase $\lambda$ and repeat the previous steps.
8: **end if**
---

of the network. Since our regularization requires sampling over the whole input domain, it requires more samples as the dimension increase, which means that the sampling could fail if the dimension of the input, $d$, is large (e.g. $d = 276$ in Blog Feedback). To address it, we add an additional linear layer for dimension reduction on the non-monotonic features. This linear layer reduces the dimension of the non-monotonic features to 10, and is also trained in an end-to-end manner.

Our method adopts a simple MLP structure. We select the number of the hidden layers (depth of the network) from $d = \{1, 3\}$ using the validation set. Since our regularization is applied on each $2k$-th layer, the number of hidden neurons is fixed to 20 to avoid curse of dimension. For the neuron numbers in each $(2k + 1)$-th hidden layer, we select from $n \in \{40, 100, 200\}$.

## B   Additional Formulation

### B.1   Individual Monotonicity for Deep Networks

For networks with more than 2 layers, we provide the corresponding MILP formulation. We follow the notations in Section 3.1. Consider the the following MLP,

$$f(\boldsymbol{x}) = \sum_{i_K=1}^{n_K} a_{i_K} \text{ReLU}(\boldsymbol{w}_K^{i_K \top} \boldsymbol{x}_K + b_K^{i_K}), \ \boldsymbol{x}_k = \text{ReLU}(\boldsymbol{w}_{k-1}^{\top} \boldsymbol{x}_{k-1} + \boldsymbol{b}_k), \ k = 1, 2, \ldots, K.$$

Here, $\boldsymbol{w}_k$ is the weight matrix of the $k$-th linear layer, and $\boldsymbol{b}_k$ is the bias. $\boldsymbol{w}_k^{i_k}$ is the $i_k$-th row of the weight matrix, and $b_k^{i_k}$ is the $i_k$-th element of the bias. Then we can replace all the ReLU activations with the linear constrains (5), and thus creating a MILP problem. Comparing with the two-layer case, we introduce a new variable $\boldsymbol{z}_k$ and the corresponding constraints in every additional layer.

### B.2   Monotonicity Verification with General Activation Function

Our method has been developed for piecewise linear functions. In this section, we extend it to to any continuous activation functions. The idea is to bound the activation function with piecewise linear functions. Specifically, consider a two-layer network,

$$f(\boldsymbol{x}) = \sum_{i=1}^{n} a_i \sigma(\boldsymbol{w}_i^{\top} \boldsymbol{x} + b_i),$$

where $\sigma$ is a general activation function. Then the partial derivative equals,

$$\partial_{x_\ell} f(\boldsymbol{x}) = \sum_{i=1}^{n} \sigma'(\boldsymbol{w}_i^{\top} \boldsymbol{x} + b_i) a_i w_{i,\ell}.$$

We can bound $\sigma'(\cdot)$ with step-wise constant functions. Assume we partition $\mathbb{R}$ into $M$ consecutive, non-overlapping intervals, such that $\mathbb{R} = \bigcup_{m=1}^{M} [p_m, q_m)$, where $q_m = p_{m+1}$, $p_1 = -\infty$, $q_M = +\infty$. Now we can bound $\sigma'(\cdot)$ with,

$$\sum_{m=1}^{M} g_m^- \, \mathbb{I}\,(x \in [p_m, q_m)) \leq \sigma'(x) \leq \sum_{m=1}^{M} g_m^+ \, \mathbb{I}\,(x \in [p_m, q_m))$$

where $g_m^- = \inf_{x \in [p_m, q_m)} \sigma'(x)$ and $g_m^+ = \sup_{x \in [p_m, q_m)} \sigma'(x)$, both of which can be calculated explicitly. If we take $M$ large enough, the upper and lower bound will approach the original $\sigma'(\cdot)$. Now we have the following lower bound for $\partial_{x_\ell} f(\boldsymbol{x})$,

$$\partial_{x_\ell} f(\boldsymbol{x}) \geq \sum_{i=1}^{n} \sum_{m=1}^{M} g_{m,i} \, \mathbb{I}\left(\boldsymbol{w}_i^\top \boldsymbol{x} + b_i \in [p_m, q_m)\right) a_i w_{i,\ell},$$

where $g_{m,i} = g_m^-$ if $a_i w_{i,\ell} \geq 0$ and $g_{m,i} = g_m^+$ if $a_i w_{i,\ell} \leq 0$. Denote,

$$U_\ell = \min_{\boldsymbol{x} \in \mathcal{X}} \sum_{i=1}^{n} \sum_{m=1}^{M} g_{m,i} \, \mathbb{I}\left(\boldsymbol{w}_i^\top \boldsymbol{x} + b_i \in [p_m, q_m)\right) a_i w_{i,\ell}.$$

Replacing $\mathbb{I}(\cdot)$ with the linear constraints in (8), $U_\ell$ becomes a MILP problem. Monotonicity is certified if $U_\ell \geq 0$.

### B.3   Naive Monotonicity Verification is Impractical on Deep Networks

Naive monotonicity verification could be problematic with deep networks. To illustrate the issue, suppose $f$ is a ReLU network with $K$ layers, with $n_k$ neurons in the $k$-th layer. Then the objective for computing $U_\ell = \min_{\boldsymbol{x} \in \mathcal{X}} \partial_{x_\ell} f(\boldsymbol{x})$ is,

$$U_\ell = \min_{\boldsymbol{x} \in \mathcal{X}} \quad \boldsymbol{a} \, diag(\boldsymbol{z}_K) \, \boldsymbol{w}_K \, diag(\boldsymbol{z}_{K-1}) \, \ldots \, \boldsymbol{w}_2 \, diag(\boldsymbol{z}_1) \, \boldsymbol{w}_1^\ell.$$

We ignore the constraints here for simplicity. Here, $\boldsymbol{a}$ is the weight matrix of the last linear layer, and $\boldsymbol{w}_1^\ell$ refers to the $\ell$-th column of the input layer $\boldsymbol{w}_1$. $\boldsymbol{z}_i = \left(z_i^1, \ldots, z_i^{n_i}\right)$ contains all the binary decision variables for the indicator functions of the $i$-th layer, where $n_i$ denotes the number of neurons in that layer. Expanding the objective leads to product of these binary variables, $z_1^{i_1} z_2^{i_2} \ldots z_K^{i_K}$, which makes the objective non-linear. We can linearize the problem by introducing new binary variables,

$$U_\ell := \min_{\boldsymbol{x} \in \mathcal{X}} \sum_{i=1}^{n} a_i \sum_{i_k \in [1:n_k]} \left[ \left( \prod_{k=1}^{K} [\boldsymbol{w}_{k-1}]_{i_{k-1}, i_k} \right) z_{i_1, i_2, \ldots, i_K} \right]$$

$$s.t. \quad z_{i_1, i_2, \ldots, i_K} \leq z_k^{i_k}, \quad z_{i_1, i_2, \ldots, i_K} \geq \sum_{k=1}^{K} z_k^{i_k} - (K-1), \quad \forall k \in \{1, 2, \ldots, K\}.$$

Here, $[\boldsymbol{w}_k]_{i_k, i_{k-1}}$ refers to the element on the $i_k$-th row and $i_{k-1}$-th column in the weight matrix $\boldsymbol{w}_k$. Intuitively, we replace the product $z_1^{i_1} z_2^{i_2} \ldots z_K^{i_K}$ with a new binary variable $z_{i_1, i_2, \ldots, i_K}$ and additional constraints to linearize the problem. However, in this way, we need $n_1 \times n_2 \times \cdots \times n_k$ new binary variables, which is an unaffordable large-scale MILP problem for typical MILP solvers. Even for a small network with 3 hidden layers and 20 neurons in each hidden layer, there is more than 800 binary variables. Current MILP solvers will fail to solve this problem in limited time (e.g. 1 hour). To summarize, naive monotonicity verification requires to consider all the paths in the neural network, which greatly increases the number of integer variables.

## C   Additional Experiment Results

### C.1   Influence of $\lambda$

$\lambda$ indicates the magnitude of our monotonicity regularization. We empirically demonstrate how $\lambda$ influence the lower bound $U_\ell$. We show 2 networks with $d = 1, n = 100$ (Net 1) and $d = 3, n = 100$ (Net 2) on COMPAS and Chest X-Ray. Generally, $\min_{\ell \in \alpha} U_\ell$ increases as $\lambda$ increases.

### C.2   Validation Results

We provide the validation accuracy of different structures on different datasets.

Figure 7: **Left**: Result on COMPAS. **Right:**Result on Chest X-Ray. Generally, $\min_{\ell \in \alpha} U_\ell$ increases as $\lambda$ increases.

| Network | Depth | Hidden Neurons | Total Parameters | Validation Accuracy |
|---------|-------|----------------|------------------|---------------------|
| 1 | 1 | 40 | 522 | 62.15% |
| 2 | 1 | 100 | 1302 | 68.02% |
| 3 | 1 | 200 | 2602 | 68.12% |
| 4 | 3 | 40 | 1792 | 65.99% |
| 5 | 3 | 100 | 7462 | 68.22% |
| 6 | 3 | 200 | 23112 | 68.42% |

Table 7: Validation Results on COMPAS

| Network | Depth | Hidden Neurons | Total Parameters | Validation RMSE |
|---------|-------|----------------|------------------|-----------------|
| 1 | 1 | 40 | 8492 | 0.1340 |
| 2 | 1 | 100 | 17192 | 0.1345 |
| 3 | 1 | 200 | 31692 | 0.1357 |
| 4 | 3 | 40 | 9762 | 0.1373 |
| 5 | 3 | 100 | 23352 | 0.1378 |
| 6 | 3 | 200 | 54002 | 0.1371 |

Table 8: Validation Results on Blog Feedback

| Network | Depth | Hidden Neurons | Total Parameters | Validation Accuracy |
|---------|-------|----------------|------------------|---------------------|
| 1 | 1 | 40 | 1082 | 64.70% |
| 2 | 1 | 100 | 2342 | 64.98% |
| 3 | 1 | 200 | 4442 | 65.03% |
| 4 | 3 | 40 | 2352 | 65.07% |
| 5 | 3 | 100 | 8502 | 65.15% |
| 6 | 3 | 200 | 26752 | 65.12% |

Table 9: Validation Results on Loan Defaulter

| Network | Depth | Hidden Neurons | Total Parameters | Validation Accuracy |
|---------|-------|----------------|------------------|---------------------|
| 1 | 1 | 40 | 5732 | 61.65% |
| 2 | 1 | 100 | 6632 | 61.76% |
| 3 | 1 | 200 | 8132 | 62.10% |
| 4 | 3 | 40 | 7002 | 60.87% |
| 5 | 3 | 100 | 12792 | 62.21% |
| 6 | 3 | 200 | 30442 | 61.20% |

Table 10: Validation Results on Chest X-Ray

## Footnotes

[4]https://archive.ics.uci.edu/ml/datasets/BlogFeedback

[5]https://www.kaggle.com/wendykan/lending-club-loan-data

[6]https://www.kaggle.com/nih-chest-xrays/sample