[Reviews · NeurIPS 2020]

Review 1

Summary and Contributions: This paper deals with learning network that are monotonous with regards to the inputs. As opposed to approaches based on achieving monotonicity by construction, the authors heurisitically encourage monotonicity at training time by penalizing negative gradients at sample points and verify that monotonicity is achieved at test time. The verification is accomplished by using standard MIP formulations of piecewise linear neural network to verify that two layer neural network are monotonous, and decomposing deep networks into a stack of two layers neural network, requiring that each two layers neural network be monotonous.

Strengths: - This paper deals with an important problem. In addition to the motivation of fairness that the authors give, enforcing monotonicity also often makes sense when we are attempting to model physical systems for which we have domain knowledge. - The contribution relies on known techniques from Neural Network verification but builds novel elements on top (such as formulating the verification objective of the gradient sign) and apply those techniques to a new setting where they haven't been applied yet. - Experimental results are convincing and clearly shown, both on small examples where visualisation is possible (Figure 2) and on quantified datasets (Table 1 to 4) where the authors obtain strong results with less parameters than other methods.

Weaknesses: The whole section 3.1 feels out of place and unnecessary at the moment. Unless I'm mistaken, if verification of global monotonicity can be achieved, there is no point in performing individual monotonicity certification. As it is, this subsection feels un-necessary and is also not empirically tested.

Correctness: The method is correct and the empirical methodology tests their training algorithm correctly. One point that would be worth discussing is the convergence of their training algorithm described in Section 3.3. Is there a possibility for the problem to never converge to a solution with positive U_\alpha, leading to an infinite loop or would there be at least some arguments for why this would not happen?

Clarity: The paper is extremely well written, and quite easy to follow. The authors do a great job at progressively building up the description of their method.

Relation to Prior Work: Yes, the paper appropriately describe its relation to both the verification literature whose methods it uses, and the "monotonous by construction" literature which it aims to improve upon. To the best of my knowledge, there is no missing discussion of related works.

Reproducibility: Yes

Additional Feedback: ====== UPDATE AFTER REBUTTAL =============== I read the reviews by the other reviewers and the rebuttal by the authors. Most of the reviews seem positive and I personally think that the combination of making verification work (by decomposing the network in 2-stack) + sampling based regularizer to encourage monotonicity is interesting, so I'm quite happy to maintain my score where it is, especially if the author include the missing related works pointed by the other reviewers. I would nevertheless advise the authors to include a rigorous justification of the comments they gave in G#2 of the rebuttal in the final version of the paper. ====== UPDATE AFTER REBUTTAL =============== l.45-46: I would be wary of claiming support for "arbitrary" neural network architecture when using MIP would restrict the supported architectures to piecewise linear one. I understand that the distinction is made with regards to special purpose monotonous architecture like DeepLattice. Section 4: Are the networks that are used and for which performance is reported all guaranteed to be monotonic over the whole input domain? If so, I would advise the authors to make a note of it. Section 4.1: Is there actually a layer with only two neurons in the architecture used?


Review 2

Summary and Contributions: The paper presents 2 interconnected methods, with the second being more significant than the first. The first is a technique for training a ReLu neural network with monotonicity constraints by augmenting the usual objective function with a term which penalizes violation of monotonicity. Although this method on its own is not as novel as it is presented as being, the second method appears to be an important and novel contribution: a method for verifying whether a ReLu neural network is exactly monotone. The overall training technique consists of a back and forth between the first method and the verification step, with the penalty for monotonicity violation being strengthened if verification fails. Verification for deep networks (more than 1 hidden layer) is addressed by decomposing a deep network into a series of 2-layer networks. Positive experimental results (superior generalization) as compared to several competing techniques are presented on several real-world problems. A method for enhancing the interpretability of more general (non-monotone) neural networks by adding a monotonic layer is also presented.

Strengths: The ability to certify exact monotonicity for the general ReLu network case (not just the trivial all-positive-weight case) is an exciting and worthwhile contribution to the literature. The experimental results are also strong. It's also encouraging to see that the method learns non-trivial monotonicity implementations, i.e. negative weights but nonetheless a monotonic function overall.

Weaknesses: I'd probably be OK with the paper being published as is, but I feel strongly that the authors should clarify one key part of their algorithm. I would be happy to increase my rating if they could make this clarification. The objective function presented in eq. 11 after line 165 is shown only at a very theoretical level which skips over important technical details about how the R(f) monotonicity-violation term is actually evaluated. As the authors themselves say, if R(f) equals 0 in the true, exact, analytic sense of the expectation w.r.t the uniform distribution over X, then the network is indeed monotonic (aside from pathological measure-0 scenarios which we will leave aside). But if it is indeed truly monotonic, then you would not need to verify it with the MILP solver. So as I tried to figure how how R(f) is implemented in practice, I deduced that it must be approximated by sampling the partial derivatives at various locations in the input space rather than being R(f) being calculated analytically. However, as far as I can tell, this sampling procedure is not described or even acknowledged anywhere in the main paper. I had to go to line 394 in the supplementary to see an acknowledgment that the regularization involves sampling. Even in the supplementary, the details of the sampling are not described. Furthermore, I need more details on the lambda multiplying R(f). If lambda is a single, real-valued parameter and the number of samples in the estimation of R(f) is fixed, then increasing lambda and retraining doesn't make that much sense to me. Obviously, you wouldn't stop training the first time unless the sampled R(f) =0, since the network is certainly not monotonic overall if the sampled R(f) > 0. Buf if the sampled R(f) does equal 0 but the certification via U_alpha fails, then I don't see why increasing lambda would necessarily help, because the parameter solution the ReLu network found the first time, which satisfied "sampled R(f)=0" and therefore lambda*sampled R(f)=0, would still be available to the network for a larger lambda and would once again achieve 0 for the regularization term and just as low a loss (L(f) ) as before on the training data. Now, I suppose in practice, with a higher lambda, the network empirically might tend to find a different local minimum wherein the network is truly globally, certifiably monotonic everywhere, but if this occurs in practice, it seems like kind of a fluky effect which I would not want to rely on in the future, since it seems very unlikely to me that the training loss component would be as good in the certifiably monotonic case as it was in the case where sampled R(f) was zero but certification fails. So is lambda just a tunable single real number and is the number of samples of the partial derivative in the empirical, sampled R(f) fixed ? Or does "lambda" really somehow represent the number of samples in the sampled R(f) ? It would make a lot more sense to me if the method involved increasing the number of samples and retraining if the certification fails, rather than increasing a real-valued lambda for a fixed sampling. Assuming I am correct that R(f) is sampled at points within the input space, how many such points were used?

Correctness: The methods are correct as far as I can tell. My main concern is the vagueness of the presentation of the regularization methodology, as previously explained. Line 63: Min/Max "does not work well in practice" seems a bit unfair given that it is probably tied with Crystal overall when looking at the 4 experiments.

Clarity: Here are several suggested edits: Line 95 This kind of problems -> This kind of problem Line 142 Following the same spirit with the precious section -> Following the same spirit as the **previous** section Line 152 abundant of heuristics -> abundant heuristics (or an abundance of heuristics) Line 157 has been an active research areas -> active research area Line 195 scarification -> increase Line 275 Since it is a fundamental machine learning methodology, We -> we

Relation to Prior Work: Training a neural net with a regularization term which penalizes violation of monotonicity on a sample of the input space was first presented in Sill/Abu-Mostafa, Monotonicity Hints, NeurIPS 1997, so this paper should be cited. Although Sill/Abu-Mostafa penalize function value differences rather than true gradients, the method is close enough that it should be cited. [13], A. Gupta et. al is not the first example of this type of technique.

Reproducibility: No

Additional Feedback: I think I would feel very good about the paper if I can get enough clarification about how R(f) is implemented in practice. **** Update after reading author feedback **** Thank you for describing the details of the R(f) implementation. After reading this explanation, I have chosen to raise my score to a 7. I *strongly recommend* that this description (512 samples, with different samples at each iteration) be included in the text of the paper itself. This is the kind of experimental detail which can make the difference between someone else successfully re-implementing the paper or not. It was very valuable for me to learn that the samples are re-generated and different at each iteration. If the samples were the same throughout training (across all iterations) then increasing lambda and re-training if certification fails would not have made any sense, since the model parameter settings resulting from the first training would have sampled R(f)=0 on those fixed samples and there would be no reason to expect the second training with increased lambda not to find that same failed parameter setting. Knowing that different samples are used at each step makes all the difference, but this info was not in the first version of the paper. So I really think that needs to be in the paper. I realize there are tight space constraints for NeurIPS papers, but there must be something else the authors can cut to fit that in.


Review 3

Summary and Contributions: In this paper, the authors proposed methods to train and verify monotonic deep learning models. The authors adopted a two-step iterative approach: first train a deep learning model with penalty on the violation of monotonicity; second, verify whether the model is monotonic through an optimization problem which can be solved using mixed integer linear programming; then, iterate to increase the penalty weight until the monotonicity is verified to be satisfied. Overall, I think this is a novel paper, with a few weaknesses. Hence I gave a score of 5.

Strengths: Training monotonic models is an important problem, with vast applications in interpretability, security and fairness. This paper proposes a method to train flexible deep learning models and are yet monotonic. The proposal is novel, and numerical studies show that the proposal is statistically on par or better than some competing methods.

Weaknesses: The biggest weakness of this work is the challenge in the optimization problem in (10). As a result, various simplifications need to make, which sacrifices the performance of the proposal. First, it is computationally intractable to verify the overall monotonicity of a deep learning model; rather, we need to decompose the deep learning model into stacks of two-layer models for verification. Second, we may need to stop MLIP early. Thus, even if no adversarial example is found, there is still no guarantee that the model is fully monotonic. Another weakness of this method lies in the numerical studies. First, the proposed method is not statistically better than DLN in Loan Defaulter or Chest X-Ray datasets. Second, the authors did not show train metrics, which made it hard to understand whether the improvement comes from more flexible modeling, or from better generalization. Third, the results are averaged over 3 runs (it is unclear what the difference is between the 3 runs), making the validity of the error bars questionable. Fourth, the author does not make it clear the specifics of the MLIP solver (error tolerance etc.). Thus, as I mentioned above, the final model may not be guaranteed monotonic, and it would be important for the author to show the final verification result of the monotonicity of the model. Finally, due the iterative nature of the proposal: train -> verify -> train ..., hyperparameter selection becomes more difficult, i.e., the smallest lambda that makes the model monotonic should also depend on the model architecture, and the performance of the model depends both on lambda and model architecture. Thus, it is unclear to me how to efficiently choose the optimal lambda and model architecture that gives us 1) a monotonic model 2) that achieves the best performance.

Correctness: The methodological aspect of the method seems correct, though I did not carefully check the math. The numerical studies have room for improvement; see above.

Clarity: The presentation of the method is clear. The presentation of the numerical studies need improvement; see above. Other comments: Numerical study results need to be expanded; see above. What are the hyperparameters used for competing methods? How did you choose them? The right panel of Figure 1 is not clear. Please elaborate. Minor points: In the product price example provided at the end of page 2, the presentation is a bit vague. I suggest the author to make this example more specific. What can this feature x be? How could adversaries utilize the attack in practice? etc.

Relation to Prior Work: The authors discussed the connection and differences with existing methods.

Reproducibility: Yes

Additional Feedback: =========================================================== I appreciate the authors to address my concerns and questions. I especially appreciate the authors to include training loss in the rebuttle. I have updated my rating accordingly.


Review 4

Summary and Contributions: The idea of monotonic networks is the ability to impose monotonicity constraints on the subset of input features. For example, it might be unfair to offer a job to X over Y when X has better test scores and all other features are similar. Most trained neural networks are not guaranteed to satisfy monotonicity constraints. To enforce monotonicity in a trained model, it requires special network architectures that can lead to restricted models. Otherwise, it is not possible to guarantee monotonicity constraints in the general networks. In this paper, the authors use MILP to detect the violation of monotonicity constraints and use appropriate monotonicity enforcing regularization to ensure that the trained model satisfies monotonicity constraints. At a high level the main idea is to keep increasing the regularization till we guarantee that the trained model satisfies monotonicity constraints.

Strengths: 1) The paper is well written and addresses an important problem in the context of enforcing monotonicity in the solution of deep neural networks. 2) The solution using MILP to check for monotonicity is elegant. 3) The experimental results are convincing and shows improvement over existing baselines on multiple datasets. 4) The application to CNNs and interpretability benefits are shown on MNIST datasets.

Weaknesses: 1) The general idea seems to use a monotonicity enforcing penalty function as the regularization term and keep increasing this till the network satisfies motonocity conditions. Is there any guarantee that we can always achieve this through training? 2) This might be a general criticism regarding the work on monotonicity enforcing networks. In many real world situations, non-monotonic features change as well. There may be cases where some monotonic features increase one way and some decrease the other way. In such cases, it is not clear as to what the proposed algorithm will do. There are more realistic cases, and the monotonicity conditions may still not be very useful. 3) The MNIST example for using monotonicity for better interpretability is interesting. It would be good to know if the monotonic network achieved better classification accuracy or not. 4) Table 5 needs better explanation. What are A,B, C, D in Table 5. Also, there should be some explanation as to why positive paths lead to monotonic solution and under what scenarios? 5) The use of consecutive 2-layer networks for enforcing monotonicity seems like a good trick to improve the computational efficiency. As the paper mentions, this may lead to a more restricted monotonic network. It would be good to know if removing this restriction at the cost of computational efficiency would lead to better performance or not. 6) I am also a bit unclear about extending this method to larger networks with CNNs or ResNet architectures.

Correctness: The claims and the method seem to be correct.

Clarity: The paper is well written except the part in Table 5 mentioned before.

Relation to Prior Work: The paper discusses the prior work. The MILP formulation used in this paper reminds me of the MILP used in the recent formulations used in Fischetti et al. 2018 and Serra et al. 2018: [a] Matteo Fischetti and Jason Jo. Deep neural networks and mixed integer linear optimization. Con- straints, 23(3):296–309, 2018. [b] Thiago Serra, Christian Tjandraatmadja, and Srikumar Ramalingam. Bounding and counting linear regions of deep neural networks. In Proceedings of the 35th International Conference on Machine Learning, ICML 2018. It would be good to discuss the MILP with respect to some of these existing formulations for DNNs, since the formulation used in the submission is only for 2 layers.

Reproducibility: Yes

Additional Feedback: I went through other reviews and the rebuttal. Overall, the authors have responded to my major concerns, and also provided the necessary clarification for Table 5.

[Author Response · NeurIPS 2020]

We thank all the reviewers. We hope the reviewers could increase the rating if the response addressed your concerns.

**G#1. General Comments: Convergence of the algorithm** Theoretically, in Eq. (11), if we take $\lambda \to \infty$ and estimate
$R(f)$ sufficiently accurate, the solution will eventually become monotonic. In practice, we found that we usually find
monotonic functions without using a lot of samples when estimating $R(f)$, and we can further manage to do so easier
by modifying $R(f) = \mathbb{E}_{x \sim Unif}[\min(b, -\partial f(x))^2]$, where $b$ is a small positive constant (Please see Appendix A2
L402-404). Accordingly, we have revised the relevant sections of the paper by adding pertinent technical details.

**G#2. General Comments:** All the learned models that we report performance about are certified monotonic.

**R1 Regarding Additional Feedback: 1. Arbitrary NN:** Our main work is for ReLU networks, extension to more
general activation functions are discussed in Appendix B2. **2. Are networks guaranteed monotonicity?** All the
networks reported are certified monotonic. We will make a note of this in the paper; see also G#1 and G#2. **3. A layer**
**with only two neurons?** You are correct that, in section 4.1, we use only two neurons for the sake of better illustration
because it is a binary classification problem.

**R2 On the Implementation of $R(f)$:** The exact value of $R(f)$ is intractable, and we approximate it by drawing
samples of size 512 uniformly from the input domain during iterations of the gradient descent. Note that the samples
we draw vary from iteration to iteration. By the theory of stochastic gradient descent, we can expect to minimize the
object function well at convergence. Also, training NNs requires more than thousands of steps, therefore the overall
size of samples can well cover the input domain. Our $\lambda$ is just a tunable real number. See also G#1. We hope you could
raise your score if this addresses your concern.

**R4** *"We need to stop MILP early. Thus, even if no adversarial example is found, there is still no guarantee that the*
*model is fully monotonic."* MILP problems are NP-hard. In our implementation, we solve the *exact solution* without
early stopping for the two-layer MILP problems, and the computation empirically only takes about several seconds
when the hidden layer has 100 neurons (Fig. 3). Moreover, most MILP solvers are based on branch and bound, which
provides a lower bound of the optimal solution and continuously tightens it. Therefore, it is in fact feasible to stop the
algorithm whenever a lower bound is larger than zero, which provides certification but requires no exact solution. See
also L149-159 for discussion.

**Regarding Numerical Studies:** (1) While achieving similar test performance, our method generally use much less
parameters than DLN. For example, in Loan Defaulter, we have 72% parameters less than DLN (see Table. 3). (2) To
address the potential issue with only 3 runs, we report in the table below the average result in 10 runs (ours) of training
loss, average test result and smallest verification result. (3) The difference of the three runs is only the random seed. (4)
We provided sensitivity analysis on $\lambda$ in Fig.6 in Appendix. Please see also the general response for more information.

| Datasets | Test Acc (DLN) | Training Loss (DLN) | Test Acc (Ours) | Training Loss (Ours) | $U_\alpha$ |
|---|---|---|---|---|---|
| COMPAS | $67.94\% \pm 0.27\%$ | 0.6124 | $68.83\% \pm 0.19\%$ | 0.5938 | 0.009 |
| Blog Feedback | $0.1608 \pm 0.0006$ | 0.1468 | $0.1584 \pm 0.0003$ | 0.1629 | 0.003 |
| Loan Defaulter | $65.12\% \pm 0.17\%$ | 0.6244 | $65.21\% \pm 0.03\%$ | 0.6235 | 0.005 |
| Chest X-Ray | $65.42\% \pm 0.13\%$ | 0.6232 | $66.07\% \pm 0.29\%$ | 0.6269 | 0.006 |

**Hyperparameters of our method:** Currently we adopt a simple strategy for choosing hyperparameters: (1) we start
from $\lambda = 1$, and multiply $\lambda$ by 10 every time the network fails to pass certification (L226-227). (2) we choose the
architecture by grid-search using the validation set (see Appendix L399-404 for choice of grid).

**Hyperparameters of Baselines: Isotonic:** Default in Scikit-learn package. **XGBoost:** Gird search parameters: maxi-
mum tree depth, $[3, 10]$; learning rate $\eta$, $[0, 1]$; sub-sampling ratio, $(0, 1]$. **Crystal and DLN:** We used hyperparameters
for the best reported results in corresponding papers.

**Right Panel of Fig. 1:** It confirms that even the network looks feature-wise monotonic, we can still find adversarial
sample $x_{adv}$ against $x$ in high dimension. Specifically, it shows $y = f(ax + (1-a)x_{adv})$ w.r.t. $a \in [-0.5, 1.5]$, with $a$
as x-axis and $y$ the y-axis. We will make that clear in future version.

**R5 Questions in "Weakness" Section:** (1) Please refer to G#1 and G#2.

(2) Thanks for pointing that out. It is beyond the scope of this work. We will investigate it in the future.

(3) The results on the three binary classification tasks: **Test Acc (Mono/Non-Mono): 2 vs. 7:** $99.42\% \pm$
$0.07\%$ / $99.53\% \pm 0.04\%$ **8 vs. 3:** $99.76\% \pm 0.06\%$ / $99.78\% \pm 0.09\%$ **7 vs.1:** $99.85\% \pm 0.08\%$ / $99.82\% \pm 0.04\%$.

(4) (1) $A, B, C, D$ refer to four different networks, with different structures, trained on different datasets. We will
include the details in the future version. (2) Let us take a two-layer ReLU network, $f = W_2 ReLU(W_1 x)$, for instance.
Because $ReLU(\cdot)$ is a monotonically increasing function, we can simply verify the monotonicity of the network if
all the elements in the matrix $W_2 W_1$ is non-negative without our MILP formulation. Each element in the matrix is a
multiplication of the weights on a path connecting the input to the output, hence named *non-negative/negative paths*.

(5) We found that directly certifying DNNs with MILP without using two-layer decomposition fails to work completely
due to the excessively high computational cost. See Appendix B.3 for more discussion.

(6) Our method can be used to verify the monotonicity on the high-level features in the last few FC layers of CNNs
(as what we do in MNIST result); it does not seem to make sense in general to assume monononicity on lower level
features such as pixels.

[Meta-Review · NeurIPS 2020]

The reviewers uniformly agreed that this is a well-written paper, on an important problem, describing a novel approach, with a good experimental evaluation. Most of their remaining concerns appear to require relatively minor changes. Of these, the most significant are (i) missing related work (R5 included some potential citations), (ii) an unclear explanation of Table 5, and (iii) that R(f) is estimated from samples, but this isn’t made clear in the main text (instead of the appendix). Overall, this is a solid paper, and a few tweaks would make it even better. Please carefully read the reviews, and take their suggestions seriously when making edits.